# Plant Defenses Against *Tetranychus urticae*: Mind the Gaps

**DOI:** 10.3390/plants9040464

**Published:** 2020-04-07

**Authors:** M. Estrella Santamaria, Ana Arnaiz, Irene Rosa-Diaz, Pablo González-Melendi, Gara Romero-Hernandez, Dairon A. Ojeda-Martinez, Alejandro Garcia, Estefania Contreras, Manuel Martinez, Isabel Diaz

**Affiliations:** 1Centro de Biotecnología y Genómica de Plantas, Instituto Nacional de Investigación y Tecnología Agraria y Alimentaria, Universidad Politécnica de Madrid, UPM, 28223 Madrid, Spain; me.santamaria@upm.es (M.E.S.); a.arnaiz@upm.es (A.A.); i.rosa@upm.es (I.R.-D.); pablo.melendi@upm.es (P.G.-M.); gara.romero@upm.es (G.R.-H.); da.ojeda@upm.es (D.A.O.-M.); alejandro.garciagar@alumnos.upm.es (A.G.); estefania.contreras@upm.es (E.C.); m.martinez@upm.es (M.M.); 2Departamento de Biotecnología-Biología Vegetal, Escuela Técnica Superior de Ingeniería Agronómica, Alimentaria y de Biosistemas, UPM, 28040 Madrid, Spain

**Keywords:** *Tetranychus urticae*, plant defense, constitutive and inducible defenses, signalling events, mite counter-defenses

## Abstract

The molecular interactions between a pest and its host plant are the consequence of an evolutionary arms race based on the perception of the phytophagous arthropod by the plant and the different strategies adopted by the pest to overcome plant triggered defenses. The complexity and the different levels of these interactions make it difficult to get a wide knowledge of the whole process. Extensive research in model species is an accurate way to progressively move forward in this direction. The two-spotted spider mite, *Tetranychus urticae* Koch has become a model species for phytophagous mites due to the development of a great number of genetic tools and a high-quality genome sequence. This review is an update of the current state of the art in the molecular interactions between the generalist pest *T. urticae* and its host plants. The knowledge of the physical and chemical constitutive defenses of the plant and the mechanisms involved in the induction of plant defenses are summarized. The molecular events produced from plant perception to the synthesis of defense compounds are detailed, with a special focus on the key steps that are little or totally uncovered by previous research.

## 1. Introduction

Plants are constantly subjected to a combination of abiotic and biotic stresses which provoke a strong selection pressure leading to plant adaptation to growth–defense trade-off. In the context of the plant interaction with pathogens and pests, plants have developed complex strategies to survive and preserve their fitness by combining constitutive and inducible defenses [1,2,3]. However, defenses against phytophagous arthropods require different strategies than those towards pathogens due to their mobility and precise feeding modes, which produce specific plant injuries. Plant responses to arthropods are based on inherited traits and are generally divided into three categories: antixenosis or deterrence, antibiosis or resistance and tolerance [4,5]. Antixenosis represents a clear nonpreference of an arthropod for a plant and is denoted by the presence of deterrent factors (colours, odors, textures) that alter arthropod behaviour and select an alternative host. Antibiosis is the plant–pest antagonistic association in which chemical and morphological defensive plant factors adversely affect the biology of the arthropod. Tolerance is the plant’s ability to interact with the arthropod and to withstand, repair or recover from the potential damage produced by the pest. These three strategies can be exclusive or can overlap mechanistically and functionally since their defensive purpose is plant specific and dependent on arthropod species and its feeding mode.

Most studies on plant–arthropod interactions have been focused in insects but mites (Chelicerata, Arachnida, Acari) also use plants as a food source, causing significant damage and yield losses. Among phytophagous acari, spider mites (Acari, Tetranychidae) are the most important family with nearly 1,300 described species in about 77 genera, of which approximately 10% are plant feeders [6]. In particular, the two-spotted spider mite *Tetranychus urticae* Koch is considered a serious threat for agriculture because it is an extremely polyphagous species with a short life cycle, high offspring production and a remarkable ability to develop pesticide resistance [7,8,9]. *T. urticae* females may produce over 100 eggs of different size depending on the embryo sex since this species reproduces by arrhenotoky, a form of parthenogenesis in which males develop from unfertilized eggs while diploid females derived from fertilized eggs [10]. Egg laying is followed by larvae hatching and the subsequent development of protonymph, deutonymph and adult stages, to complete a life cycle that takes place in less than 10 days under optimal conditions (Figure 1a).

*T. urticae* is a cosmopolitan pest that feeds on more than 1,100 documented plant species, of which 150 are important agronomic crops [11]. Besides, under the current climate change scenario associated with dry and hot conditions, *T. urticae* shortens its life cycle, produces more generations per year and broadens the host range [12]. Spider mites feed, mainly but not only, on the leaves by piercing individual mesophyll cells. They introduce a retractable stylet between epidermal cells or through the open stomata without inferring any cell damage, inject saliva to predigest the mesophyll cell content and suck its content. Consequently, the chlorophyll is lost, the photosynthetic rate is reduced and typical chlorotic lesions appear (Figure 1b,c). Under severe infestations, leaf defoliation and crop losses are produced [13,14]. When a plant host is overexploited and food resources become limited, *T. urticae* uses its ability to synthesize silk as a strategy to migrate and colonize new plants. They generate silk balls that may give refuge to hundreds of mites and are used as aerial dispersal elements spread either by wind or animal transport [15]. Besides, silk webbing over leaf surfaces protects the mite colony from external aggressions, is used for mite locomotion and acts as a pheromone substrate [16,17] (Figure 1d).

The sequence of the genome of *T. urticae* identified specific expansions of gene families involved in digestion, detoxification and transport of xenobiotic compounds to cope with defense molecules of different host plants [18]. These features support the mite polyphagous character and demonstrate its capability to counteract the effects of a variety of active components either synthesized by the plant as defenses or as components of the acaricides used for pest control. In addition, the presence of endosymbionts, mainly *Wolbachia spp*, in spider mites may alter their interaction with the host, manipulate the plant defenses and modify mite performance [19,20]. Biological control of spider mites through natural predators is a useful practice in closed greenhouses but not in open fields. Unfortunately, up to now, there are no plant cultivars with resistance to spider mites [21,22].

Working knowledge of the spider mite biology together with the development of the wide range of genomic tools and protocols established for *T. urticae* [23,24,25,26,27] make this species a model within the phytophagous chelicerate to address different aspects of the plant–spider mite interactions. To understand the underlying molecular mechanisms of plant defense against spider mites has been a major challenge in the last years [22,28]. However, plant defense responses to *T. urticae* encompass a complex and highly regulated process involving many factors, signaling molecules and pathways. In this puzzle, where all pieces are essential, there are still many knowledge gaps to understand the whole defensive landscape. This review compiles all the current knowledge of plant–spider mite *T. urticae* interplay identifying host defense mechanisms to be unveiled.

## 2. Plant Defenses Against *T. urticae*: Step by Step

### 2.1. Physical and Chemical Constitutive Barriers

The defensive response of the plant to *T. urticae* challenge includes a combination of pre-existing constitutive defenses evolutionary developed to avoid pest damage, and inducible defenses specifically generated upon spider mite infestation. There are several examples of the protective role of constitutive barriers, either physical or chemical, to combat this phytophagous acari. For instance, an increased leaf trichome density in raspberry (*Rubus idaeus* L.) correlated negatively with spider mite presence and demonstrated that this antixenosis trait hampered the spider mite movement and limited the mite egg deposition [29]. Gomez-Sanchez et al. [30] have recently shown that the thicker leaf cuticles detected in barley (*Hordeum vulgare*) *HvPap-1* knockdown plants protected leaves from *T. urticae* feeding probably by hindering stylet penetration into the leaf mesophyll. 

Regarding the chemical barriers, a clear association between glandular trichomes in wild tomato genotypes and spider mites has been described and identified as a source of resistance (Table 1). *Solanum habrochaites* Kanpp and Spooner accession LA 407 accumulates methyl ketones that had deterrence activity against spider mites [31]. *Solanum pennelli* Correll LA-716, *Solanum pimpinellifolium* TO-937 and *Solanum galapagense* Darwin and Perelata VI057400, VI045262, VI037869 and VI037239 accessions, were resistant to spider mites since the acylsucrose exudated by type IV trichomes caused mortality and reduced the oviposition rates of spider mite females [32,33,34]. Tomato genotypes rich in zingiberene, a sesquiterpene exuded by type IV and VI trichomes, presented antibiosis-type resistance by increasing nymph mortality and decreasing fecundity [35]. These defense traits make wild-tomato genotypes also resistant to other phytophagous species different than acari, and even to phytopathogens. Thus, they should be considered a source of general antibiosis-type resistant traits that could be introduced in cultivated species [36]. Bleekler et al. [37] studied an atypical terpene pathway producing high concentrations of 7-epizingiberene in wild tomato. The introduction of two genes involved in this pathway into a cultivated greenhouse variety produced the accumulation of this sesquiterpene in the transgenic tomato trichomes. The mortality of mites that fed on these tomatoes increased up to 40%; fecundity was also reduced in comparison to non-transformed cultivated tomatoes [37]. Likewise, the accumulation of other allochemicals such as flavonoids in citrus, phenolic compounds in chrysanthemum or terpenoids in cucumber and citrus have also been associated with resistance to *T. urticae* [38,39,40]. Taken together, these data indicate that constitutive barriers, whether physical (anatomical) or chemical against *T. urticae* infestation are dependent on the specific features of the host species, accession or cultivar rather than a specific response to the feeder species. 

### 2.2. Inducible Defenses: Plant Perception

Most of the plant defenses against pests, including spider mites, are induced by the herbivore, first in the tissue where the infestation takes place (local response), and then in distal undamaged tissues of the same plant (systemic response). Both local and systemic responses involve structural and chemical modifications, particularly developed against each precise feeder. Induced defenses result in both energy saving and the avoidance of self-intoxication. However, these differences entail risks due to their delay on being operational [4]. The induction of plant defenses is initiated when specific receptors (pattern recognition receptors, PRRs) detect the presence of herbivore-associated molecular patterns (HAMPs), microbe-associated molecular patterns (MAMPs) derived from herbivore and associated microbes, or damage-associated molecular patterns (DAMPs) consequence of herbivore injury. Rapidly, this set of elicitors/effectors triggers a signal transduction cascade modulated by hormones, mainly jasmonic acid (JA), salicylic acid (SA) and ethylene (ET). Subsequently, the activated pathway leads to a reprogramming of gene expression and the activation of transcription factors that regulate the synthesis of molecules with defensive properties against pests, which are recognized as direct defenses. In addition, indirect defenses, mainly volatiles, are also produced to attract natural enemies of the phytophagous arthropod or to send messages to neighboring plants [22,51,52,53]. 

Plant perception begins with the phytophagous arthropod contact. Initially, arthropod touching, plant cell disruption and deposition of chemicals from the herbivore induce a rapid and precise defensive response [54]. Thus, plant damage, pest-associated cues like insect/acari behavior, feeding vibrations, oral secretions and oviposition fluids that release specific compounds, may elicit/suppress plant defenses [2,55,56]. In the *T. urticae* case, Villarroel et al. [57] identified a battery of putative HAMPs, salivary-secreted peptides, with potential elicitor/effector function to interfere with the plant defenses. Two of these peptides, Tu28 and Tu84, suppressed mite-induced SA defenses when these were transiently expressed in *Nicotiana benthamiana* Domin plants that promoted the reproductive performance of the acari. However, the molecular role of these effector molecules is yet to be determined. Recently, two novel proteins termed tetranin 1 and 2 (Tet1 and Tet2), which are located in the salivary glands of spider mites and have been shown to induce defenses in the host plants, have been characterized [58]. The infiltration of recombinant Tet1 and Tet2 proteins in bean leaves elicited early cellular responses such as cytosolic Ca^2+^ influx, membrane depolarization and reactive oxygen species (ROS) production. Moreover, spider mite feeding assays in bean and eggplant leaves resulted in improved *T. urticae* performance due to reduced mortality and presence of fewer predatory mites. These findings showed that plants integrate specific signals and stimuli from *T. urticae* to induce defenses, but how plants modulate the simultaneous recognition of different salivary proteins/peptides requires more investigation. Nevertheless, salivary glands or oral secretions are not the only source of spider mite signals to be potentially recognized by the plant as a threat to induce defense responses. Faeces, eggs, silk and acari molts (Figure 1d–f) can be an alternative source of HAMPs to alert the plant about the spider mite attack [59]. How plant responses are induced or suppressed by these HAMPs constitutes a set of unknown events or gaps to be explored.

Despite being one of the key players in the initial steps of plant defense signalling, the identification and characterization of PRRs in the plant–herbivore interaction is largely missing. A few receptor-like-kinases with an ecto-domain involved in ligand binding, a transmembrane region and an intracellular kinase domain have been functionally characterized. They act as active participants in the recognition of HAMPs derived from phytophagous insects to trigger defense pathways. Some PRRs have been identified in plants interacting with certain lepidopteran and aphid species [60,61,62,63,64]. In contrast, the information on PRRs associated with the recognition of other insect species is very limited. 

Comparative genome-wide transcriptome analyses of plant responses to *T. urticae* have revealed the upregulation of several receptor-encoding genes either across plant species or in a single plant host [44,65]. Alysm-containing receptor-like kinase (LYK4) is involved in chitin-triggered signalling induced by mite feeding in *Arabidopsis thaliana* (thale cress), grapevine and tomato. However, its ligands/binding pairs and downstream defense signalling events have to be deciphered. Recently, Santamaria et al. [42] have characterized a spider-mite-induced thale cress gene, *PP2-A5*. This gene encodes a protein with two domains, a Toll/Interleukin-1 receptor (TIR) and a phloem protein 2 with lectin activity (PP2), located at the N- and C-terminal regions of the protein, respectively. In *PP2-A5* overexpressing or knock-out plants transcriptional reprogramming takes place leading to alter the plant defense responses against *T. urticae* that are mediated by a hormonal crosstalk. However, whether the receptor domain participates in the spider mite recognition by binding to derived-HAMPs has yet to be determined. Further investigation is required to have a clear understanding of the function of each single domain. Likewise, a recent functional characterization of a thale cress bidirectional promoter shared by two divergent genes has demonstrated the simultaneous induction of the two genes by *T. urticae* feeding. Interestingly, one of these two divergent genes encoded an LRR-receptor-kinase that could be involved in the perception of the phytophagous mites to initiate the defense transduction cascade [66]. Further investigations are needed to demonstrate its role. In conclusion, the arthropod ligand-PRR binding in the plant–spider mite interaction is one of the weakest links and presents an important knowledge gap to our understanding of the plant-spider mite interaction (Figure 2). 

### 2.3. Inducible Defenses: Early Signalling Events

The recognition of phytophagous signals, particularly those that bind to specific PRRs at the plasma membrane, produces variations in the electrochemical gradient between intra- and extra-cellular sites. Concomitant to a depolarization of the membrane potential (*V*m), the cytosolic Ca^2+^6influx and the ion channel activity increased, whereas a ROS and reactive nitrogen species (RNS) burst occurs in response to arthropod feeding. These changes take place rapidly after infestation and activate a defense-signalling cascade. Calcium sensors such as calmodulin, calcineurin, Ca-dependent protein kinases and Ca-ATPases play a crucial role to regulate downstream targets in the signal transduction cascade that eventually result in a generation of battery of defenses [2,52,67]. So far, most of the information on the early defense events comes from studies on plant–insect interaction. Also, some reports have demonstrated that *T. urticae* infestation produces cell damage that leads to alterations in cytosolic Ca^2+^ levels and *V*m variations accompanied by ROS production [58]. Accumulation of ROS and phenolic compounds at the wounding sites caused by *T. urticae* in barrelclover *(Medicago truncatula)* and thale cress plants have been reported [2,68,69,70]. ROS molecules, particularly H_2_O_2_, are essential in defense signalling and in oxidative pathways to regulate cell proliferation and promote cellular processes [71]. Excess of H_2_O_2_ causes oxidative stress and induces programmed cell death while moderate abundance of H_2_O_2_ functions as a defense-signaling molecule. Thus, ROS are essential to keep levels below damaging concentrations but triggering signals. Santamaria et al. [50] demonstrated the role of thale cress ROS homeostasis related genes that are involved in the synthesis of H_2_O_2_ and or degradation of H_2_O_2_ and ascorbate, in response to *T. urticae* infestation. Silencing of the three genes involved in ROS degradation resulted in higher leaf damage and better mite performance in comparison to wild-type plants. It has also been reported that the effect of the mite attack triggered immunity (*MATI*) gene involved in the maintenance of the thale cress antioxidant status was correlated with the spider mite performance. *MATI-*overexpressing plants led to a moderate H_2_O_2_ accumulation and increased thale cress resistance to spider mite attack as reflected by lesser leaf damage and the reduced mite fecundity observed in comparison to the symptoms and oviposition rates quantified in control and mutant plants [70]. 

There is some evidence demonstrating that ROS production depends on the activity of plant oxidases such NADPH oxidases, also known as respiratory burst oxidase homologues (RBOHs). They have the ability to integrate Ca^2+^ signalling, protein phosphorylation and ROS production as key mediators of rapid local and systemic signalling in plants in response to pathogens and pests [72,73]. *Nicotiana attenuata* Torr. ex S. Watson ROBH-silencing plants significantly reduced ROS levels and were more susceptible to the generalist lepidopteran *Spodoptera littoralis* Boisduval feeding than control plants [74]. Likewise, chemical inhibition of RBOH activity in wheat plants reduced H_2_O_2_ accumulation and altered the expression of downstream defense enzymes such as β-1,3-glucanases and peroxidases involved in wheat resistance to the aphid *Diuraphis noxia* Mordvilko ex Kurdiumov [75]. There is no known information on the role of RNS, particularly nitrogen oxide (NO), which is considered a signalling molecule in plant defense against a plethora to of pest and pathogens [76,77]. 

Another early event in response to herbivores is the activation of kinases and phospholipase C leading to the formation of phosphatidic acid, a secondary messenger involved in oxidative stress [47]. In addition, mitogen-activated protein kinases (MAPKs) are also key players of the transduction pathway. Schweighofer et al. [78] showed that the thale cress Ser/Thr phosphatase type 2C, AP2C1, acted as negative regulator of the stress-responsive MPK4 and MPK6 activities, which had a defense role not only against pathogens but also in the thale cress–spider mite interaction. Mutant *ap2c1* plants, where both kinases were highly activated, resulted more resistant to *T. urticae* than wild-type and *AP2C1* overexpressing plants. Moreover, the complementation of the *ap2c1* mutation restored the normal fecundity of spider mites [78]. 

Among the early plant responses to *T. urticae* infestation, is very important to mention the alteration of the hormone levels. Generally, JA is the main regulator of induced defenses triggered by spider mites. As Rioja et al. [28] reported, elevated JA correlates with the degree of plant resistance to *T. urticae* infestation since plants with constitutive JA-mediated responses were more resistant to spider mite populations while mutants in JA synthesis were more susceptible [44,79]. However, not only the JA pathway is activated after spider mite infestation. The induction of both JA and SA pathways has been reported in spider mite infested thale cress [42,64,70], tomato [44,47], citrus [40,43,80] and pepper [46], among other plant species. Moreover, He et al. [81] also reported the accumulation of SA and conjugated-SA forms induced by spider mite attack in bean (*Phaseolus vulgaris*) leaves but the JA content was not analyzed in this study. Although JA and SA are considered antagonist hormones, SA does not seem to antagonize JA responses in the plant–spider mite context since SA content also increases after spider mite attack. Most likely, the potential reciprocal crosstalk between JA and SA signalling pathways allows a fine-tune modulation to sense the synthesis of specific direct and indirect defensive molecules against the acari. The JA/SA dosage, their temporal dynamics and the level of activation vary among plant hosts and plant developmental stage. There are many other factors, which may have different effects on the plant defense and on spider mite behavior [82]. 

Besides JA and SA, other phytohormones participate in the signalling and regulation of plant responses against *T. urticae*. It has been demonstrated the emission of ET and other volatiles in tomato and lima bean, which are triggered by spider mite attack [44,47,83]. In addition, the abscisic acid (ABA) insensitive 4 transcription factor (ABI4) has been identified as a crucial component of chloroplast retrograde signaling that regulates mite-associated thale cress defense controlled by ABA [84]. More recently, differences have also been found in spider mite-regulated auxin levels in thale cress [42]. In addition, metabolic and hormonal pathways are shared with other environmental biotic and abiotic stimuli, and such mutual interference was reported for light stress-mites and aphid-mite interaction [84,85].

Taken together, these results suggest that the modulation of plant resistance/susceptibility to spider mites is dependent on the hormonal crosstalk but it remains to be clarified how such complex signalling cascades affect plant defenses (Figure 2). These phytohormone-mediated mechanisms are not only specific to spider mite infestation but are considered the hallmark for plant defense signalling against both biotic and abiotic stressors [86,87]. 

### 2.4. Inducible Defenses: Late Defense Events 

Early signaling triggers a series of later events that culminate to the production of compounds functioning as toxins, repellents and/or antifeedants that are involved in direct and indirect responses. Examples of plant-derived metabolites with defense properties that directly or indirectly target spider mites are presented in Table 1. 

Transcriptomic and metabolomic approaches performed on different plants species following spider mite infestation have allowed the identification of a plethora of secondary metabolites involved in defense, mainly flavonoid and terpenoid products, alkaloids derived from the shikimate pathway, and some compounds resulting of the aminobenzoate degradation. Among them, glucosinolates exclusively synthesized by Brassicaceae, were highlighted in the thale cress–spider mite relationship. In particular, indole-glucosinolates (IG), compounds derived from the tryptophan pathway, whose expression is dependent of JA, constituted the central defenses against spider mites in this model species. Zhurov et al. [41] reported that some IG metabolites were induced by mite feeding and acted as deterrents and antifeedant agents. Accumulation of IG in thale cress dramatically increased spider mite mortality. IG-deficient thale cress mutants with reduced levels of a set of IG metabolites were highly susceptible to spider mite feeding. Furthermore, the overexpression of the *ATR1* transcription factor, a positive regulator of IG gene expression made plants more resistant to spider mite infestation and increased larval mortality [41]. Although the mechanism of action of IG as antiacaricidal molecules has not been fully validated, it is well-known that thiocyanates, isothiocyanates, nitriles and other toxic compounds are derived from the glucosinolate hydrolysis mediated by myrosinases [87]. This hydrolytic process takes place in damaged tissues and the toxic products directly target the phytophagous arthropod physiology either reacting with biological nucleophiles or modifying nucleic acids and proteins [52].

Likewise, transcriptomic assays performed at different time points of mite-infested tomato plants revealed the upregulation of genes involved in the synthesis of secondary metabolites with known functions in plant defense [44,47]. Besides, the emission of volatile organic compounds, including phenolics and several classes of terpenes, was detected over a period of the first five days after infestation. The increase in the volatile production coincided with the greater olfactory preference of predatory mites for infested plants. Agut et al. [40] found flavonoid compounds such as naringenin, hesperitin and p-coumaric acid accumulated in a mite-resistant citrus genotype, and demonstrated that these metabolites were over-accumulated after mite infestation. Since some flavonoids, such as p-coumaric acid, participate in the biosynthesis of lignin polymers, their defense role was related to the formation of physical barriers to reduce the palatability of the plant. The upregulation of genes encoding flavonoids have been recently found in a comparative transcriptomic study performed in resistant common bean cultivar to spider mites [49]. In addition, macarpine, a benzophenanthridine alkaloid derived from shikimate, was also identified in mite-infested citrus [40,43]. In the same citrus genotype was detected the production of the terpene volatiles α-ocimene, α-farnesene, pinene and D-limonene, and the green leaf volatile 4-hydroxy-4-methyl-2-pentanone, all of them with a marked repellent effect on spider mites. Interestingly, volatiles released from the resistant genotype after spider mite infestation promoted mite-induced resistance in a neighbouring susceptible citrus genotype, thereby reducing oviposition rates [43]. Very recently, the combination of transcriptome and metabolome analyses in spider-mite infested pepper (*Capsicum annuum*) has allowed us to appreciate the role of terpenoids, including di- and tri-terpene glucosides, some mono-terpenes, sesquiterpenes and homoterpenes, during *T. urticae* infestation as well as alkaloids and aromatic compounds [46]. Other secondary metabolites such as O-dimethylallyleugenol, a renylated phenylpropene identified in the Japanese star anise, *Illicium anisatum* L., exhibited deterrent activity to spider mites [45]. In addition, as previously described, secondary compounds such as methyl-ketones and acyl-sugars provide chemical defense against *T. urticae* infestation [31,32,33,34,35]. These results indicate that each plant species, and even each genotype, specifically respond to spider mite infestation generating different molecules with distinct defense properties.

Besides the induction of secondary metabolites, primary defensive metabolites including proteins and peptides are also accumulated in plants after *T. urticae* infestation. The well-known example is the rapid increase of proteinase inhibitors (PIs), generally encoded by JA-dependent genes, and mainly but not exclusively found in Solanaceae plants. These proteins exert direct effects on phytophagous arthropods by interfering with their physiology, either inhibiting the gut digestive protease activities or the function of other proteases involved in growth and development [23,48]. In tomato, two serine PI genes, *PIN-I* and *PIN-II*, were shown to be consistently induced by spider mite attack [47,88,89]. Moreover, measurements of the corresponding PI activities showed about twofold increase in mite-infested leaves compared to control ones. Martel et al. [44] corroborated these results and indicated that tomato PI genes were among the most highly induced in the microarray analyses performed in the spider mite-tomato system, suggesting that they represent one of the major tomato defense response outputs upon spider mite herbivory. Likewise, the induction of two genes, At*KTI4* and At*KTI5*, encoding Kunitz trypsin inhibitors (KTIs) in spider mite-infested thale cress, and the fact that their corresponding silenced lines conferred higher susceptibility to *T. urticae* than wild-type plants, strongly support the potential defense role of KTIs [90]. In addition, barley cystatins, inhibitors of cysteine-proteases, have also been used as defense transgenes. When they were expressed alone or in combination with serine-PIs in maize and thale cress, an enhanced resistance was observed against phytophagous mites by altering mite cysteine-proteases [23,91]. In conclusion, although the list of molecules potentially involved in the plant defense against *T. urticae* is high, for most of them the mode of action and the targets in the mite remain to be remains yet to be identified. 

The mentioned early- and late-term defenses induced by the phytophagous mites can be altered by prior experiences. The earlier exposure of plants to specific stimuli may prepare it for upcoming stresses. This phenomenon termed priming, is a physiological event of immune adaptation that prepares the plant to react faster, to generate a more effective defense response with no trade-offs in the absence of challenge. A clear example of priming in plant defense to *T. urticae* has been reported by Agut et al. [43]. They demonstrated the effect of volatiles emitted from spider mite-infested sour orange and Cleopatra mandarin plants on ono-infested plants. Notably, macarpine was identified as one of the defensive compounds accumulated in response to priming treatment [43]. If the function of this compound is to induce resistance has to be proven. If so, volatiles might be used as new inducers of plant resistance to spider mites under field conditions. This hypothesis is supported by a recent publication [92] that reported how spider mites’ biological parameters, mainly oviposition on strawberry leaves, are affected by aromatic compounds derived from intercropping. Priming was also observed in the case of sequential and different herbivore infestation leading to mutual pest suppression [85]. Priming is another gap to be explored to reduce costs of plant defense.

## 3. Spider Mite Counter-Defenses

Many phytophagous organisms, and particularly the generalist *T. urticae*, have acquired traits to overcome plant defenses through three main strategies: avoidance, metabolic resistance and suppression [4,93]. The avoidance of induced plant defenses entails a behavioral feature while the other two strategies make the herbivore cope with ingested plant metabolites. Metabolic resistance against toxic molecules can arise from target-site insensitivity or detoxification mechanisms and may imply metabolite modification, degradation and/or secretion. Defense suppression is achieved via sabotage of the host plant’s molecular machinery. Thus, phytophagous mites have evolved specialized molecules secreted into or onto their host to interfere in different manners with the host’s ability to defend itself [4]. In the *T. urticae* case, the most striking example of polyphagy with a high ability to adapt to novel plants and developing rapid resistance to pesticides, the combination of the three mentioned strategies is used to maximize its performance [94]. Its own feeding mechanism may explain how spider mites try to avoid the induction of plant defenses. The stylet usually penetrates through stomata or epidermal pavement cells to reach single mesophyll cells, avoiding epidermal damage, which minimizes the detection of the attack, therefore delaying the plant response [14]. In addition, *T. urticae* genome revealed strong signatures of polyphagy linked to the expansion of gene families involved in digestion, detoxification and transport of xenobiotics. These gene families include cytochrome P450, carboxyl/cholinesterases (CCEs), glutathione S-transferases (GSTs) and ATP-binding cassette transporters (ABC). Besides, the integration of new detoxifying genes acquired by lateral gene transfer from bacteria enhances the spider mite protection to host plant toxic compounds [18,95]. However, the suppression of defenses is probably the main and most interesting strategy developed by spider mites to cope with toxicity and survive [91,94,95]. *T. urticae* and the Solanaceae-specialist *T. evansi* may suppress SA- and JA-dependent defenses in tomato plants by acting downstream of the phytohormone pathway, but each mite strain affects the expression of tomato defense genes in a different way [96]. The mite *T. evansi* can suppress JA-dependent responses by stimulating SA pathway to activate the negative crosstalk between their signalling pathways [97]. As previously commented, *T. urticae* also secretes effectors via its saliva into plant tissues to interfere with host immune response and to promote its reproductive performance [56,57]. However, the ecological costs and benefits of defense suppression are still unclear since these observations come from laboratory assays lacking the natural ecological context. In nature, mites coexist with other herbivorous and predatory mites, which may also be favored by the mite-suppression of plant defenses. Furthermore, in the case of *T. urticae* the puzzle becomes much more complicated because of the extraordinary number of genes associated with metabolic resistance, which, *a priori*, seems unnecessary to spend mite resources in the suppression of plant defenses. In fact, Blaazer et al. [93] predicted that a generalist herbivore such as this mite confined to a host for a long period of time will replace the suppression trait by resistance traits based on they are ecologically safer and promote mite performance more strongly.

## 4. Conclusions and Future Prospects

To fully understand the complex molecular interaction between a pest and its host plant is currently a major challenge for plant–pest researchers. The present compilation of the known information, regarding the molecular mechanisms involved in the arms race between the mite *T. urticae* and its host plants, highlights the great deal of key knowledge that remains unknown. In the near future, efforts should be focused on unveiling how plants perceive and signal the mite attack. To that end, it is necessary to determine the receptors of the plant that play a role in the interaction as well as the molecules of the mite that bind and form complexes with plant molecules. Besides, it will be essential to establish the consequences of these interactions, if the mite molecules act as elicitors and trigger plant immune response, or as effectors by hampering plant defense. Broad transcriptome, proteome and metabolome analysis of the responses to the mite in different plant species, as well as similar experimental approaches related to the mites feeding on them, would provide a wide core of molecular data to establish working hypotheses. Then, genetic modifications of plants and mites could be done to check these hypotheses and to fill in the existing gaps in the molecular mechanisms that control the plant–mite interplay. An advanced understanding of the mite–plant interaction will be a strong tool to enhance crop performance by improving specific plant defenses against *T. urticae* attack.

## Figures and Tables

**Figure 1 plants-09-00464-f001:**
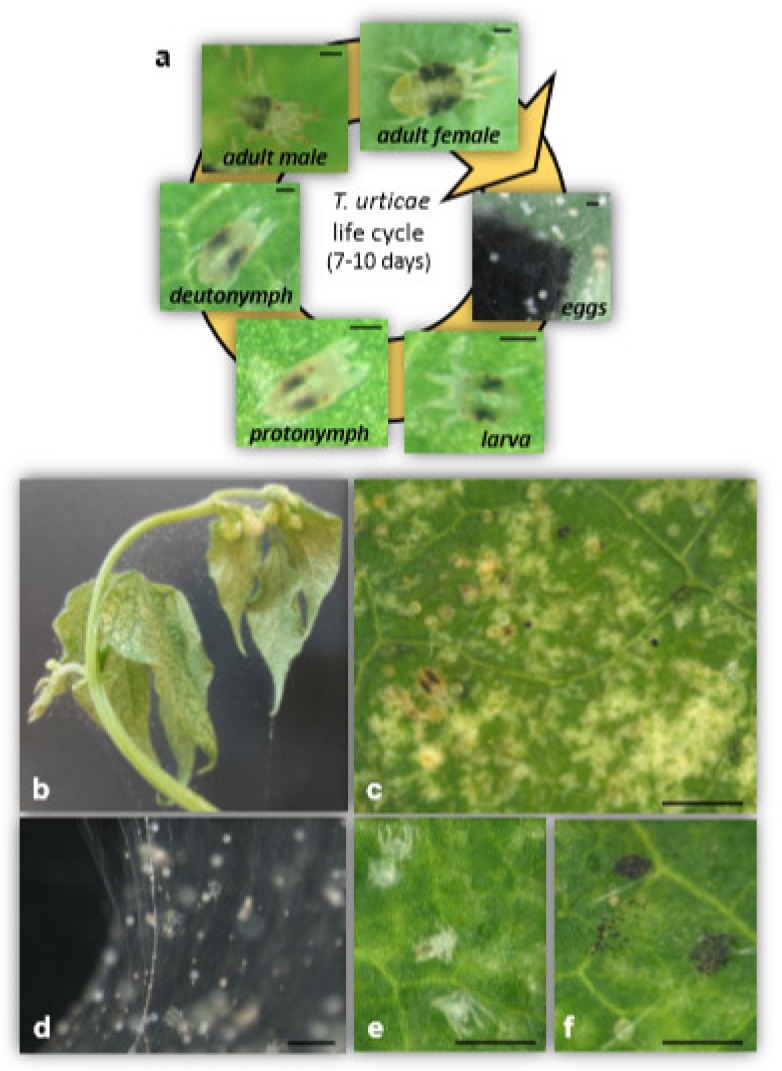
Phenotypic features of the plant–spider mite interaction. *T. urticae* life cycle (**a**), symptoms of mite-infested bean plants (**b**,**c**), and spider mite silk web (**d**), molts (**e**) and feces (**f**). Bar scales are indicated: 100 µM (**a**), 500 µM (**c**), 200 µM (**d**) and 250 µM (**e**,**f**).

**Figure 2 plants-09-00464-f002:**
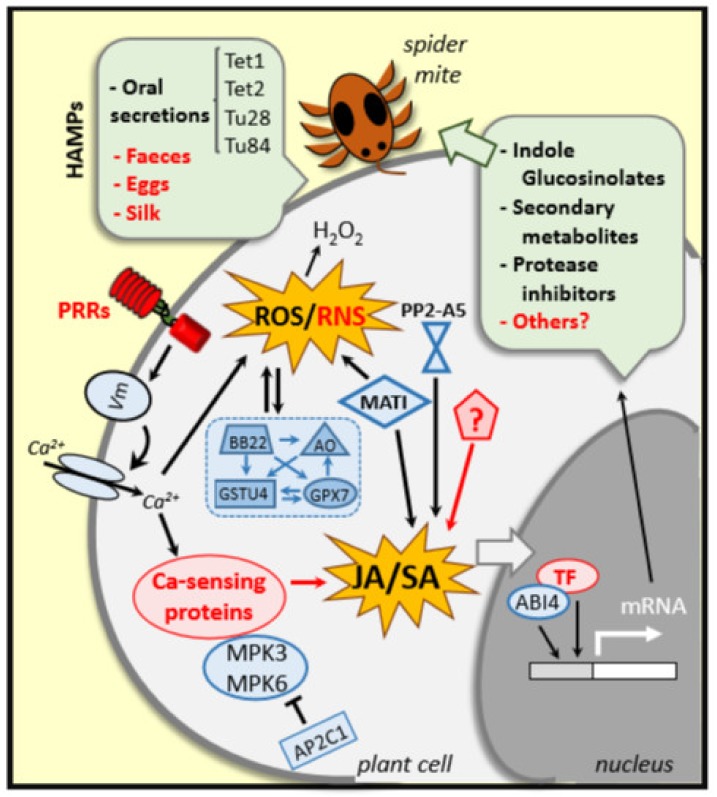
Plant event in responses to *T. urticae* infestation. Specific plant receptors (PRRs) recognize elicitors/effectors (HAMPs) derived from either the plant or the spider mite that induce alterations in the membrane potential (*V*m), cytosolic Ca^2+^ influxes and ROS/RNS burst. Ca^2+^-sensing proteins, MPKs and phosphatases (APC21) participate in the defense transduction pathway. H_2_O_2_ content is highly regulated by ROS-related enzymes (BB22, AO, GSTU and CPX7). Besides, genes such as *MATI*, *PP2A5* and others still unknown participate in the tight regulation of the hormonal crosstalk, mainly in the Jasmonic Acid/Salicylic Acid balance. All together plus some transcription factors (ABI4 and other unknown TFs) regulate the induction of the synthesis of a battery of defense molecules. Unknown genes, molecules, pathways and responses are indicated in red.

**Table 1 plants-09-00464-t001:** Plant-derived molecules with defense properties directly targeted to *T. urticae.*

Molecules	Effects on Spider Mites	Plants	Reference
Indole-glucosinolates	Toxicity	Thale cress	Zhurov et al. 2014 [41] Santamaria et al. 2019 [42]
Flavonoids	Palatability/Toxicity	Chrysanthemum	Kielkiewicz & van de Vrie 1990 [38]
		Citrus	Agut et al. 2014, 2015 [40,43]
		Tomato	Martel et al. 2015 [44]
		Common bean	Hoseinzadeh et al. 2020 [45]
		pepper	Zhang et al. 2020 [46]
Phenyl propanoids	Toxicity	Thale cress	Zhurov et al. 2014 [41]
		Tomato	Martel et al. 2015 [44]
Terpenoids	Repellence/Toxicity	Citrus	Agut et a. 2015 [43]
		Tomato	Kant et al. 2004 [47]
			Bleeker et al. 2012 [37]
			Martel et al. 2015 [44]
			Oliveira et al., 2018 [35]
		Cucumber	Balkema-Boomstra et al. 2003 [39]
		Pepper	Zhang et al. 2020 [46]
Alkaloids	Deterrence	Citrus	Agut et al. 2014, 2015 [40,43]
		Pepper	Zhang et al. 2020 [46]
Phenolics		Tomato	Kant eta al. 2004 [47]
			Martel et al. 2015 [44]
Acyl-sugars	Repellence	Tomato	Alba eta l. 2009 [32]
			Lucini et al. 2015 [33]
			Rakha et al. 2017 [34]
Methyl ketones	Deterrence	Tomato	Antonious & Snyder 2015 [31]
Phenolics	Toxicity	Star anise	Koeduka et al. 2014 [48]
Protease inhibitors	Atinutritive	Tomato	Li et al. 2002 [49] Kant et al. 2004 [47]
			Martel et al. 2015 [44]
		Thale cress	Santamaria et al. 2018 [50]

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
