# Peer review of "Plant Defenses Against Tetranychus urticae: Mind the Gaps"

_plants, 2020, doi:10.3390/plants9040464_

Round 1
Reviewer 1 Report
Very nice review. It will be helpful for many plant-mite interaction researchers. Congratulations! I enclose only a few corrections and suggestions. Please consider them.
line 58 - figure 1 - I’m not convinced whether the final picture quality will be satisfactory. Please provide better picture quality.
line 78 - the redundant comma after cited reference 18
line 105 - Table 1 and whole text - in the table there is "arabidopsis", in the text "Arabidopsis" or "A. thaliana" (without the first usage in a full form) - I propose to use: "thale cress" instead of "arabidopsis" or "Arabidopsis" instead of "Arabidopsis" or "A. thaliana" with the first usage "Arabidopsis thaliana ..."
line 152 - "study" instead of "studied"
line 153 - "a couple" or "two"
line 163 - "acari seedlings" ???
line 219 - "mediate" or similar instead of "sense"
line 255 - instead of "short-plant" use "early plant"
line 269-270 - The activation of ROS-dependent, metabolic and hormonal pathways are shared with other environmental biotic and abiotic stimuli, and such mutual interference was reported for light stress-mites and aphid-mites interaction (e.g.:
Barczak-Brzyżek AK, Kiełkiewicz M, Gawroński P, Kot K, Filipecki M, Karpińska B (2017) Cross-talk between high light stress and plant defence to the two-spotted spider mite in Arabidopsis thaliana. Exp Appl Acarol. 2017 Oct;73 (2): 177-189. doi: 10.1007/s10493-017-0187-x
Kiełkiewicz M, Barczak-Brzyżek A, Karpińska B, Filipecki M. (2019) Unravelling the Complexity of Plant Defense Induced by a Simultaneous and Sequential Mite and Aphid Infestation. Int J Mol Sci. 2019 Feb 13;20(4). pii: E806. doi: 10.3390/ijms20040806)
- line 353 - do you mean early and late defense? moreover phytophagous is an adjective – do you mean „phytophagous mites”?
- line 365 - Priming was also observed in the case of sequential and different herbivore infestation leading to mutual pest suppression (e.g.
Kiełkiewicz M, Barczak-Brzyżek A, Karpińska B, Filipecki M. (2019) Unravelling the Complexity of Plant Defense Induced by a Simultaneous and Sequential Mite and Aphid Infestation. Int J Mol Sci. 2019 Feb 13;20(4). pii: E806. doi: 10.3390/ijms20040806).
Author Response
We appreciate very much all reviewer´s comments and corrections. According to his/her indications, we have modified, and hopefully enhanced, the manuscript content.
-Figures will be sent in a different format to increase tehir quality.
-All minor mistakes have been corrected.
-The term "thale cree" is now used in Table 1 and in the whole text instead of other terms such as Arabidopsis, or A. thaliana. Following the same criteria, besides the Medicago truncatula name is mentioned te term barrelclover.
-A sentence, supported by the reference (Kielliewcz et al. 2019), has been included to mention the effect mediated by environmental abiotic and biotic stimuli and their interference in plant defence to spider mites. In consequence, reference numbering has been altered. Thus, bibiography has been reumbered.
Reviewer 2 Report
This is review article describing the different components of spider mite-host plant interaction with a focus towards understanding the genetic resources that can be used to complement traditional agricultural practices to ensure effective spider mite control.
The manuscript address the various host plant defenses that are well documented not only during spider mite interactions but also during several other insect herbivore and plant pathogen interactions. In addition the authors identify the exisiting knowledge gaps that need to be filled to gain a better understanding of the plant-spider mite interactions.
The concepts and ideas addressed in this manuscript are intersting and pertinent within the domain of plant insect interactions. The main weakness of this review article is the poor writing style, which make it difficult for the authors to convey the concepts with ease. These are specific comments and suggestions that need to be addressed by the authors:
- Figure 1a, c, e and f.The image quality is very poor in these figures. Please submit images of enhanced dpi (300 and above). Please refer to the specific journal instructions for figures. Figure legend says "seedlings"....what do authors refer to the nymphs?
- line 85 change to "working knowledge".
- line 87 change to "interactions".
- line 88 change to "underlying molecular mechanisms of plant defense against spider mites'
- line 90 change to '.....mny factors, signallng molecules and pathways.."
- line 91 change to "knowledge gaps".
- line 96 change "englobes" to "include"
- line 103 "knock -down plants? Please elaborate...knock down for which gene(s)? Correlates which phenotypes to the genotypes (knock downs)?
- line 107 Table 1 which citation?
- line 103 Include a table for physical barries in host plants that provide defense against spider mites.
- line 116 change to "other phytophagous species"
- line 117 change "resistance genes"to "resistant traits
- line 122 change to "......40% fecund and ...."
- line 126 change to "constitutive barrieers, whether physical (anatomical) or chemical against T.utricae
- line 129 & 132 changer feeder to herbivore or insect pest.
- line 132 change "defence induction" to "induced defenses
- line 137 Can all the microbes be endosymbionts? suggested change "herbivore and associated microbes'.
- line 142 change '......defense properties against pests, which are recognized as direct defenses."
- line 145 change to "induces"
- line 144 changes "phytophagous" to "phytophagous mites/acari". Please make this edit throughout the manuscript.
- line 150 change to "when these were transciently..."
- line 151 change to ".....plant that promoted the reproductive preference of acari".
- line 152 delete "this....." Suggested edit "However the molecular role of these effector molecules are yet to be determined."
- line 153 delete "a couple"
- line 157 change to "Agro-infiltrate"
- line 158 change to “….resulted in improved T.utricae performance due to reduced mortality and presence of fewer predatory mites.”
- Line 162 change to ‘…..plant as a threat to induce defense responses…”
- Line 163 Acari seedlings? Please explain
- Line 164 change to “How plants responses are induced or suppressed..”
- Line 173 change to “species and …….are very limited”
- Line 176 change to “A LYSM…..
- Line 178 change to “…….and downstream defense signaling events have to….”
- Line 179 change to “…..have characterized a spider mite induced Arabidopsis gene PP2-A5. This gene encodes a protein that ……”
- Line 182 change to “In PP2 overexpressing or knockout plants transcriptional reprogramming takes place leading to altered plant defense responses against T.utricae that are mediated by hormonal cross talk.”
- Line 184 change to “however, whether the ……yet to be determined”
- Line 185 change to “Further investigation is required to have a clear understanding of the function of each domain”
- Line 190 please change Phytophagous here and throughout the manuscript.
- Line 192 change to “……one of the weakest links and presents an important knowledge gap to our understanding of the plant spider mite interactions”
- Line 205 delete cell
- Line 206 change sides to sites
- Line 208 change “…..burst occurs in response to acari feeding ..”
- Line 208 Change to “ These changes takes place rapidly after infestation and activates a defense signaling cascade.”
- Line 211 change “batter” to “suite”
- Line 214 change to “Accumulation of ROS….A. thaliana plant have been reported (2,61-63).
- Line 216 change to “ROS molecules”. Rephrase the entire sentence
- Line 218 change to “……while moderate abundance of H2O2 functions as a defence signialling molecule.”
- Line 219 change to “Thus, ROS ……is essential to keep…”
- Line 220 change to “………..Arabidopsis ROS homeostasis related genes that are involved in synthesis of H2O2 and or degradation of H2O2 and ascorbate in response ot T.utricae infestation.”
- Line 224 Rephrase “In this scenario….”
- Line 241 change to “there are no known information on the role of RNS, particularly NO which is considered important signaling molecule in plant defense responses against a plethora of pest and pathogens (60,71).”
- Line 244 delete
- Line 255 Rephrase the sentence. In the current form it is not easy to understand what the authors want to convey.
- Line 257 Is elevated JA correlated to the resistance?
- Line 261 change to “…….spider mite infested Arabidopsis…..”
- Line 265 JA and SA are not antagonists during spider mite infestation? Please elaborate.
- Line 281 Change to “These phytohormone-mediated mechanisms are not only specific to spider mite infestations but are considered the hallmark for plant defense singalling against both biotic and abitoc stressors (citations).
- Line 283 change to “……..triggers a series of later events that culminates to the production of compounds functioning as toxins, repellents, deterrents, and/or antifeedants that involved in direct and indirect defense responses.
- Line 292 change to “tryptophan pathway” and “compounds”.
- Line 295 delete agents. Delete “The”. Suggested change, begin the sentence with “Accumulation of IG…”
- Line 303 Edit phytophagous
- Line 307 change to “metabolites with known functions in plant defense”
- Line 307-8 Rephrase sentence.
- Lne 323 change to “……has allowed us to appreciate the role of terpenoids ……..during T.uticae infestation”
- Line 326 change to “other secondary metabolites such as O-………exhibited deterrant activity specifically against spider mites (84).”
- Line 330 delete “defensive role”. Suggested edit “chemical defense against T.utricae infesations…’
- Line 334 delete “best”
- Line 339 Poor word choice. Suggested to change “retrieved” to “observed”.
- Line 352 Please change “elucidate yet” to “remains yet to be identified”
- Line 354 please change “previous exposition” to “earlier exposure of plants to specific stimuli…”
- Line 358 rephrase sentence
- Line 387 change to “cope with…”
Author Response
Er appreciate very much reviewer´s comments, particularly the list of corrections to improve the writing style. Thanks a lot for this. According to his/her indications, we have corrected all mentioned errors and suggestions.
We consider that a new table with examples of the plant physical barriers against T. urticae does not provide additional information, because most of the physical barriers are related to trichomes, and many of these examples correspond to glandular trichomes already indicated in Table 1.
Figures have been uploaded in a different format to enhance their quality and their legends have also been corrected.
Two new references have been included, as reviewer indicated, highlighting the importance of phytohormones in plant responses to abiotic and biotic stresses. In consequence, reference numbering has changes. Bibliography hsa been renumbered.
Reviewer 3 Report
Dear Authors,
I found the review excellent. Complete, punctual and very informative. A work that can be the basis of study for many researchers who want to approach this area of research. Thanks for the good job!
There are only few corrections or suggestions.

Author Response
We appreciate very much reviewer´s comments and corrections. According to his/her indications, we have corrected all minor erros.
Round 2
Reviewer 2 Report
Thank you considering the suggested edits to the manuscript. This has definitely improved the writing style immensely.
The figure quality is also much better than the fisrt submission.
However there are still quite few typos and minor spelling errors. Please review the manuscript thoroughly before submitting.
Author Response
Thank you for your comments. The whole manuscripy has been corrected to imporve the writing style.